# Immunotherapeutic Strategies for Head and Neck Squamous Cell Carcinoma (HNSCC): Current Perspectives and Future Prospects

**DOI:** 10.3390/vaccines10081272

**Published:** 2022-08-07

**Authors:** Lei Gao, Anqi Zhang, Fuyuan Yang, Wei Du

**Affiliations:** 1The First Affiliated Hospital of Yangtze University, Jingzhou 434020, China; 2Huabei Petroleum Administration Bureau General Hospital, Renqiu 062550, China; 3Department of Physiology, School of Basic Medicine, Yangtze University Health Science Center, Jingzhou 434020, China; 4Department of Oncology, The First Affiliated Hospital of Yangtze University, Jingzhou 434020, China

**Keywords:** head and neck squamous cell carcinoma, neoantigen, vaccine, adoptive cell therapy, immunotherapy

## Abstract

Neoantigens are abnormal proteins produced by genetic mutations in somatic cells. Because tumour neoantigens are expressed only in tumour cells and have immunogenicity, they may represent specific targets for precision immunotherapy. With the reduction in sequencing cost, continuous advances in artificial intelligence technology and an increased understanding of tumour immunity, neoantigen vaccines and adoptive cell therapy (ACT) targeting neoantigens have become research hotspots. Approximately 900,000 patients worldwide are diagnosed with head and neck squamous cell carcinoma (HNSCC) each year. Due to its high mutagenicity and abundant lymphocyte infiltration, HNSCC naturally generates a variety of potential new antigen targets that may be used for HNSCC immunotherapies. Currently, the main immunotherapy for HNSCC is use of immune checkpoint inhibitors(ICIs). Neoantigen vaccines and adoptive cell therapy targeting neoantigens are extensions of immunotherapy for HNSCC, and a large number of early clinical trials are underway in combination with immune checkpoint inhibitors for the treatment of recurrent or metastatic head and neck squamous cell carcinoma (R/M HNSCC). In this paper, we review recent neoantigen vaccine trials related to the treatment of HNSCC, introduce adoptive cell therapy targeting neoantigens, and propose a potential treatment for HNSCC. The clinical application of immune checkpoint inhibitor therapy and its combination with neoantigen vaccines in the treatment of HNSCC are summarized, and the prospect of using neoantigen to treat HNSCC is discussed and proposed.

## 1. Introduction

Head and neck squamous cell carcinoma (HNSCC) is currently recognized as the 7th most common cancer worldwide. Globally, an estimated 900,000 new cases are diagnosed each year, and morbidity and mortality rates continue to increase; the rates are projected to increase by 30% by 2030 [1,2]. Most head and neck cancers are head and neck squamous cell carcinomas (HNSCCs), which originate in the epithelium of the mouth, oropharynx, larynx, and lower pharynx. They are often caused by high-risk human papillomavirus (HPV) infection and adverse lifestyle factors, such as smoking, chewing betel nut and heavy drinking; other head and neck cancer risk factors include radiation exposure, vitamin deficiencies, periodontal disease, immunosuppression, and other environmental and occupational exposures [3,4,5,6]. To date, of the more than 220 HPV subtypes identified, 19 high-risk HPV types (16, 18, 26, 31, 33, 35, 39, 45, 51, 52, 53, 56, 58, 59, 66, 68, 70, 73 and 82) have been detected in the head and neck, of which HPV16 is the most common tumorigenicity subtype. Approximately 80% of HNSCC cases worldwide are HPV16 positive. Other high-risk HPV genotypes, such as HPV18, 31, or 33, are also causative but are less common [7,8,9]. Compared with early HNSCC, which can be treated with surgery or radiotherapy with a high cure rate, locally advanced HNSCC is treated with radiotherapy, chemotherapy, and targeted and other multimodal combination therapies but still has a high recurrence rate (50%) and a poor prognosis, with an average overall survival (OS) < 1 year. The 5-year OS rate of patients with HNSCC who received traditional treatment strategies, such as surgery, radiotherapy and chemotherapy, is only 30–65% [10,11]. Therefore, the development of new treatment methods is urgently needed.

In the past ten years, with the development of immunotherapy, clinical antitumor therapy strategies have changed. Immune checkpoint inhibitor (ICI) therapy has become the main method of antitumor therapy. Its main mechanism is to block signalling pathways inhibiting immune checkpoints and to reactivate the immune response to tumour cells (Figure 1) [12,13]. Related ICIs have been shown to be effective treatments for patients with R/M HNSCC. However, they are only effective in a subset of patients with R/M HNSCC due to barriers to using antitumor immunotherapy, such as primary and secondary drug resistance. Therefore, questions remain about how to improve immunotherapy for R/M HNSCC [14,15]. HNSCC naturally generates a variety of potential neoantigens due to its high degree of mutagenicity and frequent lymphocyte infiltration [16]. Neoantigens contribute to tumour-specific immune responses and have been used to develop new tumour vaccines or cellular immunotherapies [17,18].

## 2. Advantages of Targeting Neoantigens

Genetic instability of tumour cells often leads to a large number of mutations [19]. Mutations are categorized according to the consequences: (1) encoded amino acids remain the same, while the change in the DNA sequence is a synonymous mutation; (2) the mutation alters the translated amino acid, resulting in a change in the amino acid sequence of the protein that is defined as a nonsynonymous mutation; (3) nonsense mutation refers to a change in the gene sequence that changes the original amino acid into a stop codon, which stops translation; and (4) a frame-shift mutation is an increase or decrease in the nontriple number of bases in the DNA chain that will cause the original codon to shift, altering all amino acids after the mutation position. Tumours evolve through the constant generation of mutations. On the other hand, the expression of nonsynonymous mutations may produce tumour-specific antigens, and the mutation-generated antigens precisely trigger T cell antitumour immunity [20,21].

Somatic mutations alter the amino acid coding sequences, causing tumour cells to express abnormal proteins that are not found in normal cells. In cells (tumour cells or antigen-presenting cells (APCs)), these abnormal proteins are degraded into short peptides (antigen epitopes) with high affinity for major histocompatibility complex (MHC) class I or MHC class II molecules, and are presented on the cell surface in the form of complexes. These complexes are identified as targets by T cells, causing T cell activation and promoting T cell-mediated attack and removal of tumour cells. These abnormal proteins that induce T cell activation are known as “neoantigens”, and mutations capable of producing neoantigens are immunogenic mutations (Figure 2) [22,23].

Neoantigens are a class of tumour-specific antigens that are different from traditional tumour-associated antigens (TAAs). TAAs are not unique to tumour tissue because they are also present in normal tissue. TAAs are expressed at high levels in proliferating tumour cells; for example, tumour cells express *HER2*, *MUC1* and *MAGE* [24]. Currently, most cancer vaccines focus on TAAs, which are usually overexpressed. However, as normal autoantigens in the body, TAAs have two major problems in clinical application. 1: The existence of autoantigens in the body may cause the immune system to produce central tolerance. Central tolerance refers to the tolerance formed by T and B cells encountering autoantigens during the development process from the embryonic to postnatal periods. In other words, the immune system does not treat these autoantigens as foreign bodies and therefore does not produce a strong antitumor immune response. 2: Under certain conditions, if a strong immune response against tumour-related antigens is induced, autoimmune abnormalities may be caused, and immune-mediated killing effects on normal tissues of the human body may also be produced [25]. However, tumour neoantigens are only expressed in tumour cells and do not exist in normal cells, indicating that they are a foreign product relative to the human body. Therefore, compared with TAAs, tumour neoantigens are not affected by central immune tolerance, may induce a strong and persistent antitumour immune response, and have strong affinity for MHC molecules and T cell receptors (TCRs) [26]. Neoantigen-targeted therapies are safe and do not damage normal tissues and organs. As neoantigens are not expressed in normal tissues and have high immunogenicity because they are the immune products of tumour cell gene mutations, they activate CD4+ and CD8+ T cells and have strong immune activity; thus, neoantigens have been used as new targets for new precise and personalized tumour immunotherapies [22,27].

Whether a gene mutation can generate a new tumour antigen depends on (1) whether the gene mutation sequence is converted into a protein; (2) whether the mutated protein is processed into polypeptides; (3) the affinity between MHC molecules and the mutant peptide; and (4) the binding of TCRs to the mutant peptide-MHC complex [28,29,30].

Advances in next-generation sequencing (NGS) and bioinformatics technology have been achieved. NGS technology has enabled the rapid comparison of tumour and normal somatic cell genome sequences [31]. We are able to clearly understand all the mutations that produce abnormal proteins and determine which peptides have a high affinity for MHC molecules to identify neoantigens. These strategies can also detect how likely these peptides are to be recognized by T cells and monitor dynamic mutation changes in combination with liquid biopsy (ctDNA/CTC) to predict drug resistance [32,33].

## 3. Identification and Screening of Neoantigens

Because mutations usually occur in random locations, including both noncoding and nonsense mutations, neoantigens exhibit a high degree of individual specificity [34]. The identification of neoantigens is usually performed using high-throughput sequencing techniques to rapidly compare the DNA sequences of tumour cells and normal cells [35]. Because mutations in tumour cells are complex, screening for mutated proteins from these mutated DNA sequences is a challenge. The identification and screening of neoantigens are also important for cancer vaccine therapy. Unlike prophylactic vaccines, cancer vaccines are administered to patients with cancer through targeted injections and with antitumor treatments, such as combination therapies, to activate the patients’ own immune responses and kill tumour cells. Neoantigens are presented by MHC molecules and subsequently trigger an antitumor immune response in the body [36,37].

MHC class I neoantigens are usually screened according to their immunogenicity, which mainly includes the two situations described below. (1) The mutant peptide directly interacts with TCRs, suggesting that TCRs distinguish the mutant peptide from the corresponding wild-type (WT) peptide. At this point, the mutant peptide is considered exogenous and induces a T cell response [38]. (2) The mutated sites of the mutant peptide increase the strength of MHC class I binding; this enables the more efficient expression of the peptide-MHC complex on the cell surface, and T cells then recognize tumour cells as invading agents of the body and produce a response [39]. In addition to the two cases listed above, mutant peptides that bind to MHC class I molecules in a manner that sequesters them from the TCR interaction but does not alter the binding of the mutant peptide to MHC I may be considered endogenous rather than immunogenic and therefore have no immune value [40,41].

With the increasing understanding of tumour immunity, studies have revealed that CD4+ T cells activated only by recognizing MHC class II antigen and activating the immune response of helper T cells play an important role in the antitumour activity of CD8+ effector T cells. When the two T cells cooperate, they activate a sufficiently powerful antitumor immune response [42,43]. Currently, CD4+ T cells and MHC class II molecule restricted antigenic peptides have received increasing attention in tumour immunity.

The advantages of the CD4+ T cell response mediated by MHC class II-restricted antigens are described below. (1) The activation of CD4+ T cells is independent of antigen presentation by tumour cells, and its presentation mechanism is often influenced by the regulation of MHC II and costimulatory factors. (2) Most of the nonsynonymous mutations in tumours are recognized by CD4+T cells rather than CD8+ T cells, suggesting that MHC II-restricted neoantigens may be more abundant targets for immunotherapy. (3) The immune response mediated by CD4+ T cells may reshape the tumour microenvironment (TME), and the tumour invasion ability of CD4+ T cells and CD8+ T cells is significantly enhanced, resulting in myeloid-derived suppressor cells. MDSC(s) and forkhead box P3 (FoxP3) + T cells may be significantly reduced [44,45].

In addition, due to the role of CD4+ T cells in antitumor immunity, MHC II molecule- restricted neoantigens are expected to stimulate more powerful and lasting immune responses alone or in conjunction with MHC class I molecule restricted antigens [46]. In addition, 80–90% of immunogenic neoantigen epitopes screened from three different mouse models of melanoma, colon cancer and breast cancer are recognized by CD4+ T cells rather than CD8+ T cells, suggesting that tumour-specific neoantigens may bind more easily to MHC class II molecules than to MHC class I molecules with more restrictive characteristics [47,48,49]. In conclusion, MHC class II molecule-restricted neoantigens may be a promising immunotherapy target.

MHC type II immunogenic neoantigens have been screened using the following methods: (1) whole-exome sequencing (WES) or RNA-seq in tumour tissues to identify nonsynonymous somatic mutations; (2) synthesis of trimethylguanosine and antigenic polypeptides containing nonsynonymous mutations to activate autologous APCs; and (3) coculturing autologous CD4+ T cells with APCs, after which neoantigen and neoantigen-specific T cells are identified by monitoring the negative transcriptional regulation of cytokines such as IFN-γ, CD137, OX-40 and PD-1 (Figure 3).

## 4. HNSCC Neoantigen Therapy

At present, the development and application of HNSCC tumour neoantigens mainly include two forms of tumour vaccines and adoptive immune cell therapy, as well as some forms of combined application with conventional cancer therapy and ICIs. The preparation process of neoantigen vaccine is shown in Figure 4.

The principle of tumour vaccine therapy is to introduce tumour antigens into patients in the form of tumour cells, tumour-associated proteins or peptides, and genes expressing tumour antigens to overcome tumour-induced immunosuppression and enhance immunogenicity as a method to activate the patient’s own immune system and induce cellular and humoral immune responses that control and eliminate tumours [50,51]. For example, in an HNSCC preclinical study of a syngeneic mouse oral cancer cell line, a combination of NGS data and an electronic analysis detected neoantigen candidates. Validation of candidate vaccines showed that prophylactic vaccination with SLP derived from the mutant ICAM1 neoantigen induced significant T cell responses and significant tumour suppression. Given the relatively high tumour mutation burden (TMB) of HNSCC, cancer vaccine therapy with neoantigen-derived SLPs is a promising approach for HNSCC treatment [52,53]. The neoantigen vaccine mechanism is shown in Figure 5.

Adoptive cellular immunotherapy involves collecting human autoimmune cells and culturing them in vitro to increase their number by a thousand times, thus increasing their targeted killing capacity, and then transfusing them back into patients to kill the cancer cells and mutant cells in the blood and tissue. Adoptive cell therapies (ACTs) include natural killer (NK) cell therapy, tumour-infiltrating lymphocyte (TIL) therapy, chimeric antigen receptor (CAR) T cell therapy, and engineered T cell receptor (TCR) T therapy [54,55,56,57,58].

### 4.1. HNSCC Vaccination Studies

HNSCC is generally divided into HPV(+) and HPV(−) types [59]. Antitumor vaccine selection is also different for different types of patients. Studies have shown that the HPV vaccine is safe and effective at preventing HPV-associated cervical lesions [60], and thus the HPV treatment vaccine may be an effective treatment for patients with HPV(+) HNSCC [61]. Erminia et al. are conducting a phase II clinical trial of a combination of immune checkpoint inhibitors and tumour-specific vaccines for recurrent HPV-driven cancer. The trial enrolled 24 patients (22 with HNSCC, 1 with anal cancer, and 1 with cervical cancer). Using nivolumab in combination with the HPV-16 vaccine, the overall response rate was 33%, and the median OS was 17.5 months. Compared with patients treated with PD-1 inhibitors alone, the HPV-16 vaccine resulted in a significant increase in the production of specific T cells, but no significant increase in the ORR, which was considered due to immunosuppression [62]. To date, few clinically useful biomarkers are available for HNSCC. A small percentage of patients with R/M HNSCC exhibit sustained responses and prolonged survival after treatment with PD-1 or PD-L1 inhibitors. PD-L1-, PD-L2- and IFN-γ-related gene markers are potential immune biomarkers, but their clinical application has yet to be proven. The use of IHC to detect the HPV status of P16 in oropharyngeal tumours or directly testing HPV DNA or RNA to identify clinically and biologically distinct HPV-positive HNSCC subsets is currently the only clinically useful prognostic biomarker.

Patients with HNSCC usually carry multiple high-frequency mutations, such as TP53, CDKN2A, PTEN, PIK3CA and HRA, as well as some recurrent mutations, such as those in the NOTCH, JAK-STAT and other signalling pathways. Mutations involving the tumour suppressor genes TP53 and CDNK2A are the most common. In patients with HNSCC, mutations are more frequently detected in the tumours of smokers than in nonsmokers. An analysis of The Cancer Genome Atlas (TCGA) data showed that up to 20% of tumours (55/279) in the TCGA-HNSCC cohort may carry mutations in other genes involved in components of the antigen-presenting mechanism, including TAP1, TAP2, LMP2, LMP7, LMP10 and calreticulin. TCGA identified a subgroup of oral tumours associated with favourable outcomes that were characterized by an increased frequency of HRA and PIK3CA activating mutations and inactivating mutations of CASP8, NOTCH1, and TP53 [63]. CUL3, NFE2L2 and KEAP1 are common targets for HPV-negative HNSCC mutations. Mutations in genes including HLA-A, HLA-B, B2M and TGFBR2 may lead to defects in the immune monitoring of tumours. For patients with HPV(−) HNSCC, cancer vaccines targeting high-frequency mutations may improve antitumor efficacy. A phase I clinical trial (NCT03313778) of the personalized vaccine mRNA-4157 in solid tumour is currently ongoing. Data from previously published clinical trials showed an overall response rate of 50% in 10 enrolled patients with HPV-negative HNSCC; 2 achieved complete responses and 3 achieved partial responses. Meanwhile, the disease was stable in 4 patients, and the disease control rate (DCR) reached 90%. Although the number of cases is small, the positive effect of personalized vaccines on patients with HPV-negative HNSCC is anticipated [64].

A systematic search was conducted to identify clinical trials related to neoantigen vaccines for HNSCC. As some of the keywords of the trials were the drug names and they did not neoantigen vaccines or personalized vaccines, 5 phase I/II clinical trials of HNSCC neoantigen vaccines were obtained after screening. As shown in Table 1 (NCT03633110, NCT05269381, and NCT04266730 included patients with other tumours in addition to patients with HNSCC, as represented by other patients in the table), these trials were designed to explore the safety and immunogenicity of neoantigen vaccines or as adjuvant interventions to radiotherapy, chemotherapy, and other immunotherapy methods. Among them, NCT01998542 was conducted earlier, and it has been approximately ten years since its early publication; therefore, it is no longer reviewed. The remaining four trials are also ongoing, and some of the results are described below.

The sample size of current trials is small for the following reasons: 1. most neoantigens are derived from unique mutations that are not shared between different patients, requiring personalized treatment; 2. the cost of preparing neoantigen vaccines is high and unaffordable for many families, and 3. the vaccine preparation period is long. For patients with advanced tumours, the survival period is usually short, and an excessively long vaccine preparation period may cause patients to miss the opportunity for treatment.

The recently concluded ASCO General Assembly presented the latest data on the neoantigen vaccine TG4050 for ovarian and head and neck cancers. Eight patients were treated: 4 with recurrent ovarian cancer, 3 with complete remission of HNSCC, and 1 with recurrent HNSCC. Preliminary data suggest that TG4050 is safe, well tolerated, and induces T cell responses, regardless of the HLA haplotype [65].

We should also be concerned that many cancer vaccine trials targeting neoantigens have failed in patients with HNSCC and other cancers. The cause may be defective antigen targets (such as antigens with low immunogenicity, low specificity, or low expression levels on tumour cells) or weak adjuvants [66]. These failures may also mean that vaccines alone cannot overcome barriers to anti-tumor immunity or tolerance. It is encouraging to note that several clinical trials have demonstrated the potential for synergies between neoantigen vaccines and ICIs [67,68,69]. Clinical trials of neoantigen vaccines in combination with ICIs for HNSCC are currently underway, and are reviewed below.

### 4.2. HNSCC ACT

Tumour adoptive cell therapy is designed to induce, activate and amplify the progenitor cells of its own or allogeneic antitumor effector cells using activators such as IL-2, an anti-CD3 monoclonal antibody and a specific polypeptide in vitro and then transfer them to patients with tumours to improve the antitumor activity of patients, achieving the purpose of treatment and prevention of recurrence [70,71]. The advantage of this therapy is that the effect is fast and the effect of internal factors is relatively small [72].

#### 4.2.1. CAR-T-Cell Therapy

CAR-T therapy is designed to extract immune T cells from a patient and genetically program them in vitro to recognize specific gene segments that kill tumour cells and enhance antitumor efficacy. The CAR T cells are grown in large numbers in the lab, and then the amplified “enhanced” immune T cells are injected back into the patient to kill the tumour cells [73].

Currently, the major success of CAR T therapy has been achieved against haematologic tumours [70,74], with only partial progression observed for solid tumours [75,76]. Using MUC1 as a target in HNSCC, Mei et al. found that CAR-MUC1-IL22T cells exerted an effective cytotoxic effect on MUC1+HNSCC cells. CAR-T cells have some potential as a treatment for HNSCC [77].

Park et al. analysed the characteristics of HNSCC in TCGA and identified nine candidate CAR-T antigens among previously reported CAR-T antigens, ultimately selecting the antigen CD70 to investigate the function of its CAR-T cells in the specific recognition and killing of HNSCC in vitro. CD70 was expressed at high levels in 4 (19%) of the 21 tumour biopsies and was intensely expressed on the surface of tumour cells in 3 of the 4 samples. CD70-specific CAR T cells were generated and further shown to be effective in recognizing and killing CD70-positive HNSCC cells but not CD70-negative cancer cells. Based on these data, CD70 is clearly not an appropriate CAR-T cell target for all HNSCCs. However, for 10–20% of patients with HNSCC presenting high CD70 expression, CAR-T cell therapy may be an appropriate and attractive option [78].

Wendell Lim et al. implanted a receptor system called synNotch into CAR T cells, which not only targets solid tumour cells by recognizing proteins specifically expressed on solid tumour cells but also has the ability to find and kill other nearby cancer cells. In experiments, a single intravenous infusion of Synnetch CAR T cells targeting EGFRvIII (a neoantigen detected in several tumours) in mice with glioblastoma produced higher antitumour efficacy and T cell durability than conventional CAR T cells. No external tumour killing effect was observed [79,80]. Carl June et al. investigated CAR T cell therapy targeting prostate-specific membrane antigen (PSMA) and further optimized CAR T cells to overexpress dominant-negative TGFβRII (TGFβRDN). Thus, TGF-β signalling was blocked to enhance antitumor immunity in patients. Data from this phase I trial (NCT03089203) show that PSM-targeting CAR T cell therapy for solid tumours is feasible and generally safe [81,82]. The results of the CAR T cell-targeted HNSCC tumour-associated antigen test and other studies using CAR T cell-targeted neoantigens in other tumours suggest that the search for novel antigens in HNSCC may provide new prospects for CAR T therapy for this tumour.

#### 4.2.2. NK-Cell Therapy

NK cell therapy is when autologous or allogeneic immune effector cells are isolated and NK cells induced in vitro are delivered back to patients with tumours, allowing the cells to directly or indirectly kill tumour cells in the body and achieve the purpose of tumour treatment [83].

NK cells are an important component of the innate immune system and one of the important components of the first line of defence against pathogen invasion and malignant tumours [84]. In HNSCC, NK cells play an important role in tumour monitoring and control, and the degree of NK cell infiltration is closely related to the prognosis. Adoptive NK cells are obtained from a wide range of sources, including donor or autologous peripheral blood, umbilical cord blood, human embryonic stem cells, human induced pluripotent stem cells (hiPSCs), and NK cell lines [85].

The high-affinity NK (haNK) cell therapy developed by Friedman et al. has the ability to directly kill HPV-positive and -negative HNSCC cell lines under different experimental conditions [86]. Lim et al. used NK cells in combination with cetuximab to treat recurrent/metastatic nasopharyngeal carcinoma in a phase I clinical trial. No intolerable side effects were observed with NK cell transfusions. Among 7 subjects, 4 achieved stable disease and 3 progressed, suggesting that the combination therapy showed some efficacy [87].

NK cells used for adoptive cell therapy can be transplanted into new environments with different MHC expression patterns without losing their function [88,89]. Compared with T lymphocytes, NK cells do not induce graft-versus-host disease and play a regulatory role in most cases [90]. With the development of genetic modification technologies, NK cells have been further tailored, including the introduction of CARs and knockout of inhibitory genes [91]. Theresa E Schnalzger et al. documented the strong and highly specific activity of CAR NK-92 cells against organic-like EGFRv III transduction through luciferase measurements [92]. Chia-ing Jan et al. developed a CAR NK cell therapy targeting human leukocyte antigen G (HLA-G). In vitro, HLA-G CAR NK cells have been shown to effectively induce the lysis of breast, brain, pancreatic, and ovarian cancer cells, reduce xenograft growth and extend the median survival in orthotopic mouse models [93]. Therefore, we postulate that the search for neoantigens in HNSCC may provide a new direction for CAR NK cell treatment of HNSCC.

#### 4.2.3. TCR-T-Cell Therapy

TCR-T therapy begins by screening the TCR sequence that specifically binds to the target antigen in the body and then inserting a transgene encoding this sequence into the patient’s peripheral blood-derived T cells (or heterologous T cells) through genetic engineering. The modified T cells are then transfused back into the patient to allow them to specifically recognize and kill the antigen-expressing tumour cells, thus achieving the goal of killing the tumour [94,95]. HNSCC is usually highly immunosuppressive, and the main reasons are described below. (1) In the TME of patients with head and neck tumours, tumour-associated immune cell subtypes are often missing, their numbers are reduced, or a large number of functionally impaired immune effector cells (tumour-infiltrating cytotoxic T cells, natural killer cells, tumour-associated macrophages and myeloid suppressor cells, etc.) are present. (2) Immunosuppressive cytokines produced by tumour cells, such as vascular endothelial growth factor, transforming growth factor β, interleukin (IL-6), IL-10, and granulocyte-macrophage colony stimulating factor, and immunosuppressive cell populations produce local and systemic immunosuppressive effects. (3) Human leukocyte antigen class I expression and antigen processing and presentation mechanisms on the surface of head and neck tumour cells are defective. (4) TME factors, such as hypoxia and abnormal vascular and lymphatic proliferation, may also affect cytokine release and immune cell recruitment [9,96]. Approaches targeting the low number of T cells in the tumour microenvironment and the use of tumour-specific TCR-engineered T cells that recognize cancer-specific neoantigens may represent promising strategies for the treatment of HNSCC.

Wei et al. used blood-derived T cells to replace tumour-infiltrating lymphocytes (TILs) and HLA-matched APC lines to replace autologous dendritic cells in their experiments. TCR was isolated from reactive TIL cultures, and its function was tested using TCR-T cells in vitro and in vivo. Small gene structures that stimulate the strongest TIL response can be identified to determine the targeted neoantigens. TCR-T cells were activated and showed cytotoxicity. Specific TCR-T cells destroy human tumours in mice. Although this treatment does not completely eradicate tumours, it inhibits tumour growth, providing theoretical support for further clinical trials [97].

## 5. HNSCC Immunotherapy

The immune system of the body functions in immune surveillance. When tumour cells invade, the immune system recognizes and removes them according to the tumour antigens expressed on their surface, thus preventing the development of tumours [98]. However, in some cases, tumour cells evade immune surveillance through a variety of mechanisms, including malignant proliferation and tumour formation. The genesis and development of tumours are closely related to the TME, which is based on the blood circulation and lymphatic system, interacts with surrounding cells, and then affects the progression of tumours [99,100]. The TME is a cellular environment in which tumours or tumour stem cells exist. It is mainly composed of surrounding immune cells, blood vessels, extracellular matrix, fibroblasts, lymphocytes, bone marrow-derived inflammatory cells, signalling molecules, and excessive soluble mediators [101]. Adaptive immune cells (T and B lymphocytes) and innate immune cells (macrophages, neutrophils, mast cells, bone marrow cells, natural killer cells and dendritic cells, etc.) exist in the **TME** and interact with tumour cells through direct contact or signals transmitted by chemokines and cytokines. The subsequent evolution and development of these cells determines the course of the tumour and influences its treatment [102,103].

TAMs are innate immune cells that have been divided into M1-type macrophages with antitumour effects and M2-type macrophages with pro-tumour effects. Type M2 is dominant in the TME of HNSCC and is closely associated with adverse reactions. The expression of PD-1 on TAMs is negatively correlated with the phagocytic ability of antitumor cells. In vivo, blocking PD-1-PD-L1 might increase the phagocytic function of macrophages and reduce tumour growth. Therefore, PD-1-PD-L1 therapy may also work through a direct action on macrophages, which may have important implications for the design of ICI drugs or their combination with neoantigens in the treatment of R/M HNSCC [104].

Currently, with the in-depth research and rapid development of cancer molecular biology and immunology, immunotherapy as a tumour treatment has shown great prospects for development [105]. Studies have identified a number of “immune detection sites” that positively or negatively regulate the effectors of cancer immunotherapy. Immunodetection of programmed cell death receptors and their ligands suggests that they play an important role in tumorigenesis, development and immune escape [106,107]. ICIs or their combination with neoantigen therapy for the treatment of R/M HNSCC are new therapies that are administered following surgery, radiotherapy, and chemotherapy.

### 5.1. PD-1 Checkpoint Inhibitor Drug

PD-1 checkpoint receptors expressed on T cells interact with the ligands PD-L1 and PD-L2 on cancer cells and immune-infiltrating cells to promote immunosuppression. Therefore, blocking the interaction between PD-1 and PD-L1 or PD-L2 may promote the reactivation of the immune system, resulting in long-lasting antitumor effects on patients [108,109]. Clinical PD-1-targeted drugs for HNSCC currently include pembrolizumab and nivolumab [110], which are highly specific humanized IgG4 monoclonal antibodies that inhibit PD-1. They are currently used in patients with R/M HNSCC following disease progression after treatment with platinum-containing chemotherapy drugs [111].

#### 5.1.1. Nivolumab

Nivolumab was the first PD-1 monoclonal antibody approved by the Food and Drug Administration (FDA) for tumour therapy [112]. Ferris et al. conducted a trial comparing nivolumab with standard therapy for recurrent HNSCC (NCT02105636). They randomly divided 361 adult patients with R/M HNSCC into an experimental group (*n* = 240) that was administered 3 mg/kg nivolumab intravenously every two weeks and a control group (*n* = 121), that was administered traditional standard quasi-therapy (methotrexate, docetaxel, or cetuximab). The median survival was 7.5 months in the nivolumab group and 5.1 months in the standard treatment group. The median PFS was 2.0 months in the experimental group (19.7% achieved 6-month PFS and 36% achieved 1-year PFS) and 2.3 months in the control group (9.9% achieved 6-month PFS and 16.6% achieved 1-year PFS). Grade 3 or 4 treatment-related adverse events occurred in 31 patients (13.1%) in the experimental group and 39 patients (35.1%) in the standard care group. Nivolumab extends the overall survival of patients with platinum-refractory, relapsing HNSCC. Based on the results of trial NCT02105636, the FDA approved nivolumab as a treatment for platinum-resistant HNSCC. Ferris et al. followed patients enrolled in trial NCT02105636 for 2 years. Compared with the standard treatment group, nivolumab treatment group experienced a longer OS benefit, and treatment safety was good during the follow-up period, with no new safety problems identified [113,114].

#### 5.1.2. Pembrolizumab

The PD-1 inhibitor pembrolizumab has the same mechanism of action and is very similar in structure to nivolumab. Binding of this drug to the PD-1 receptor leads to a conformational change in PD-1 that prevents it from binding to PD-L1, thus allowing the continuous activation and proliferation of T cells and promoting the elimination of tumour cells [115,116].

Seiwert et al. administered pembrolizumab to 60 patients with PD-L1-positive R/MHNSCC. The PD-L1 expression rate was ≥1% in all patients, including 23 HPV-positive patients (38%) and 37 HPV-negative patients (32%). Patients were observed to determine their response to a 10 mg/kg pembrolizumab intravenous infusion every two weeks for 24 months. The median follow-up was 14 months (interquartile range (IQR) 4–14 months). The overall response rate (ORR) was 18% (25% for HPV+ patients and 14% for HPV− patients). The median progression-free survival and OS were 2 months and 13 months, respectively [117]. According to this study, pembrolizumab was well tolerated and had clinically significant antitumor activity against R/MHNSCC, providing a clinical basis for further research into pembrolizumab as a treatment for HNSCC.

Cohen et al. compared pembrolizumab monotherapy in patients with R/M HNSCC with methotrexate, docetaxel, or cetuximab (NCT02252042). Two hundred forty-six patients were enrolled in the pembrolizumab group, and 234 were enrolled in the control group. The experimental group received 200 mg of pembrolizumab intravenously every three weeks. The control group was administered intravenous methotrexate 40 mg/m^2^ weekly, intravenous docetaxel 75 mg/m^2^ every three weeks, or intravenous cetuximab 250 mg/m^2^ weekly. The median OS of the standard care group was 6.9 months, and 8.4 months in the pembrolizumab group. Thirty-three patients (13%) in the pembrolizumab group and 85 patients (36%) in the control group experienced grade 3 or higher treatment-related side effects. The preliminary results suggest that pembrolizumab is clinically effective and has a good safety profile in patients with R/M HNSCC [118].

#### 5.1.3. PD-1-Targeting Therapy Combined with Neoantigen Therapy

GEN-009 is a personalized neoantigen vaccine consisting of 4–20 amino acid peptides that was discovered and selected by the Atlas platform, a biometric assay that recognizes neoantigens.

Currently, a multicentre phase I/II a study (NCT03633110) is being conducted in patients with five cancers, including HNSCC. Previously published data showed that the GEN-009 vaccine was well tolerated by nine patients with solid tumours (the number of patients with HNSCC was not specified) with only discomfort at the local injection site reported. In vitro, vaccination produced an immune response against 94% of the administered peptides, as well as CD8+ and CD4+ immune responses. Currently, GEN-009 is being tested in combination with PD-1 inhibitors [119].

### 5.2. PD-L1 Checkpoint Inhibitor Drugs

Similar to PD-1, PD-L1 inhibitors block the immunosuppression mediated by PD-L1 binding to PD-1 on the surface of T cells and reactivate T cells to kill tumour cells, thereby inhibiting tumour growth. In contrast to PD-1 inhibitors, PD-L1 inhibitors are IgG1 antibodies, that have the biological function of recognizing pathogenic antigens and recognizing PD-L1 expressed on the surface of tumour cells. Second, PD-L1 binds to ligands other than PD-1. CD80 is also an important ligand of PD-L1 [120]. CD80 is an important ligand of CD28, and the interaction of CD80 and CD28 transmits the activation signal to T cells. In addition to blocking the PD-1-PD-L1 pathway, PD-L1 inhibitors block the ability of PD-L1 on the tumour cell surface to bind CD80 on the T cell surface. The release of CD80 binds it to CD28 on the surface of T cells and contributes to the long-term maintenance of the antitumor activity of T cells. Finally, PD-L1 inhibitors do not block the PD-1-PD-L2 pathway, preserving the function of macrophage PD-L2 signalling and avoiding the occurrence of side effects [121]. Currently, clinical PD-L1 drugs for HNSCC include durvalumab and atezolizumab [110].

#### 5.2.1. Durvalumab

Durvalumab blocks the binding of PD-L1 and PD-1 to CD80 to prevent tumour immune escape, activates and releases T cells, and induces an immune response. Segal et al. selected 62 patients with HNSCC and evaluated the safety and efficacy of durvalumab. Durvalumab was administered to the patients at a dose of 10 mg/kg every 2 weeks for 12 months; when disease progression or unacceptable toxicity occurred, the medication was discontinued. The incidence of adverse drug-related reactions and grade III and IV adverse events were 59.7% and 9.7%, respectively. The ORR was 6.5%, the median response rate was 2.7 months, and the median OS was 8.4 months. The trial revealed the manageable safety and encouraging durable antitumor effects of durvalumab on patients with HNSCC [122].

Ferris et al. conducted a phase III clinical trial. In patients with R/M HNSCC, the authors compared the function of durvalumab or durvalumab plus tremelimumab to the standard of care (SoC). Significant differences in OS were not observed. The 12-month survival rates of the three groups were 37.0%, 30.4% and 30.5%, respectively. The incidence of grade III and higher treatment-related adverse events was 10.1%, 16.3% and 24.2%, respectively. Although neither durvalumab nor durvalumab + tremelimumab showed a statistically significant OS benefit compared with the SoC group, the differences in the 12-month survival rate indicated the activity of durvalumab in the clinical treatment of patients with HNSCC [123].

#### 5.2.2. Atezolizumab

Atezolizumab is a humanized IgG1 monoclonal antibody targeting PD-L1. In a phase I clinical trial, 32 patients with head and neck cancer (HNC) were enrolled, of whom 21 (66%) experienced adverse events and 4 (13%) experienced grade 3–4 adverse events, with a median PFS of 2.6 months and median OS of 6.0 months. This trial proved that atezolizumab has tolerable safety and certain clinical activity [124]. However, due to the small sample size, the mechanism of action and clinical efficacy of atezolizumab against HNSCC requires further exploration.

#### 5.2.3. PD-L1 Inhibitors Combined with Neoantigen Therapy

RO7198457 is a personalized RNA-Lipoplex neoantigen vaccine that encodes up to 20 neoantigens. In a phase Ib study (NCT03289962), RO7198457 was administered in combination therapy with atezolizumab to 132 patients with advanced solid tumours, including head and neck cancer carcinoma. The T cell response to neoantigen was detected in 77% of patients in vitro. Vaccine-specific CD8+ T cells were detected in peripheral blood with a frequency of >5%, and vaccine-specific TCR was also detected in tumour specimens after vaccination [125].

At present, immune checkpoint inhibitors have achieved good results in the treatment of HNSCC [126]. We summarize the advantages and disadvantages of ICIs (Table 2). In terms of combined neoantigen therapy, we propose that more high-level clinical trials and more abundant clinical data will help us gradually improve the choice of comprehensive treatments to achieve the maximum benefit for patients.

## 6. Conclusions

With the reduction in the sequencing cost, the continuous advances in artificial intelligence technology and the deepening understanding of tumour immune mechanisms, dynamic tracking of the whole process of tumour development in patients and capturing the clonal diversity of tumour-related somatic mutations have become possible. At present, many problems and challenges persist in the neoantigen therapy of HNSCC, such as the high specificity of neoantigens, which leads to a series of practical problems, such as the long period and high cost of preparation of neoantigen vaccines or other biological agents targeting neoantigens. Attempts will be made to identify common neoantigens in patients with HNSCC as a method to provide rapid and effective treatment to patients with specific gene mutations. The next research direction will be to unify and standardize the neoantigen prediction algorithm and improve the accuracy of neoantigen prediction. At the same time, a single neoantigen may facilitate tumour immune escape and the development of immune tolerance, and may even lead to tumour recurrence. Further studies are needed to determine whether multiepitope vaccines will cover a wider range of tumour cells and reduce the possibility of immune escape.

At present, the combination of various therapeutic methods is the most effective method to fight tumours. Based on the mechanism of action of the tumour therapeutic vaccine itself, which has low biological toxicity and good safety, the combination of the tumour therapeutic vaccine with a variety of treatment methods may reduce the incidence of adverse reactions, exert synergistic antitumor effects, delay and control tumour progression, and provide greater clinical benefits than a single treatment method. More patients will benefit from personalized antitumour therapy based on neoantigen vaccines.

Overall, neoantigen vaccines for cancer may be the next preferred combination treatment for achieving long-term efficacy in cancer therapy. With the in-depth understanding of the mechanism of tumour and immune interactions, the continuous performance of in-depth neoantigen prediction research and the accumulation of validation databases, as well as the continuous attempts to test more HNSCC neoantigen-related therapies in phase I/II clinical trials, neoantigen vaccines will transition to clinical practice, initiating a new era of precision neoantigen vaccine therapy.

## Figures and Tables

**Figure 1 vaccines-10-01272-f001:**
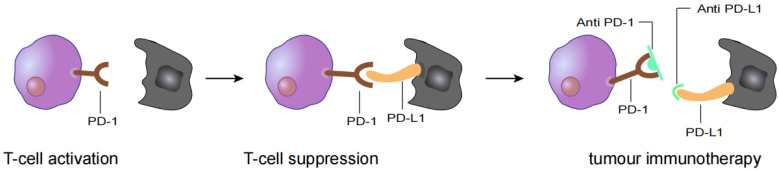
T cell activation is suppressed by the interaction between programmed death 1 (PD-1) on T cells and PD-L1 on tumour cells. Antibody drugs for cancer immunotherapy bind to PD-1 or PD-L1, blocking the PD-1/PD-L1 interaction.

**Figure 2 vaccines-10-01272-f002:**
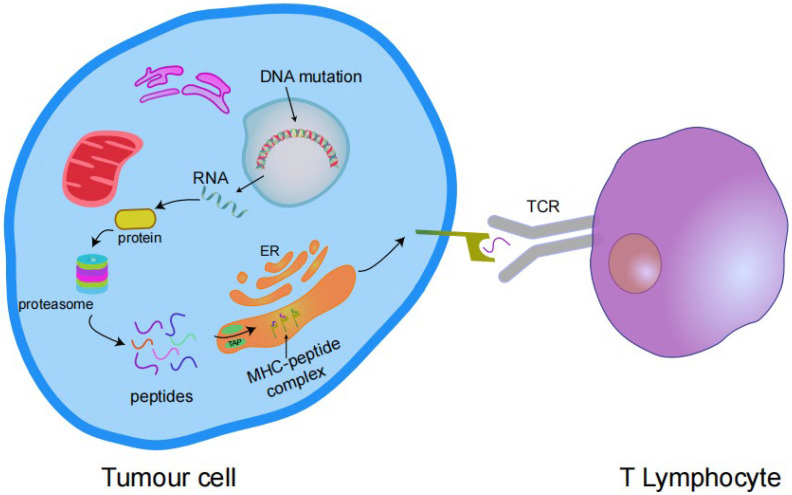
Transcription of mutant genes results in the expression of antigenic proteins that are processed by the proteasome into peptides and enter the endoplasmic reticulum (ER) via transporters associated with antigen processing (TAP) complexes. These peptides bind to the MHC I complex and are expressed on the cell surface and then bind to the T cell receptor (TCR) to activate T cells.

**Figure 3 vaccines-10-01272-f003:**
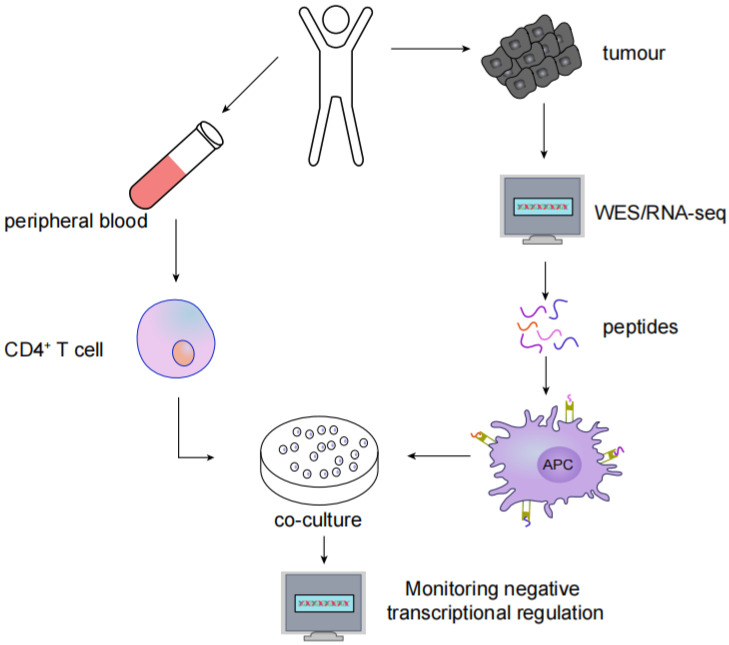
Screening for MHC type II immunogenic neoantigens: (1) WES or RNA-seq in tumour tissues to identify nonsynonymous somatic mutations; (2) Synthesis of antigenic peptides containing nonsynonymous mutations to activate autologous APCs; (3) Autologous CD4+ T cells are cocultured with APCs; (4) Neoantigen and neoantigen-specific T cells are identified by monitoring the negative transcriptional regulation of cytokines.

**Figure 4 vaccines-10-01272-f004:**
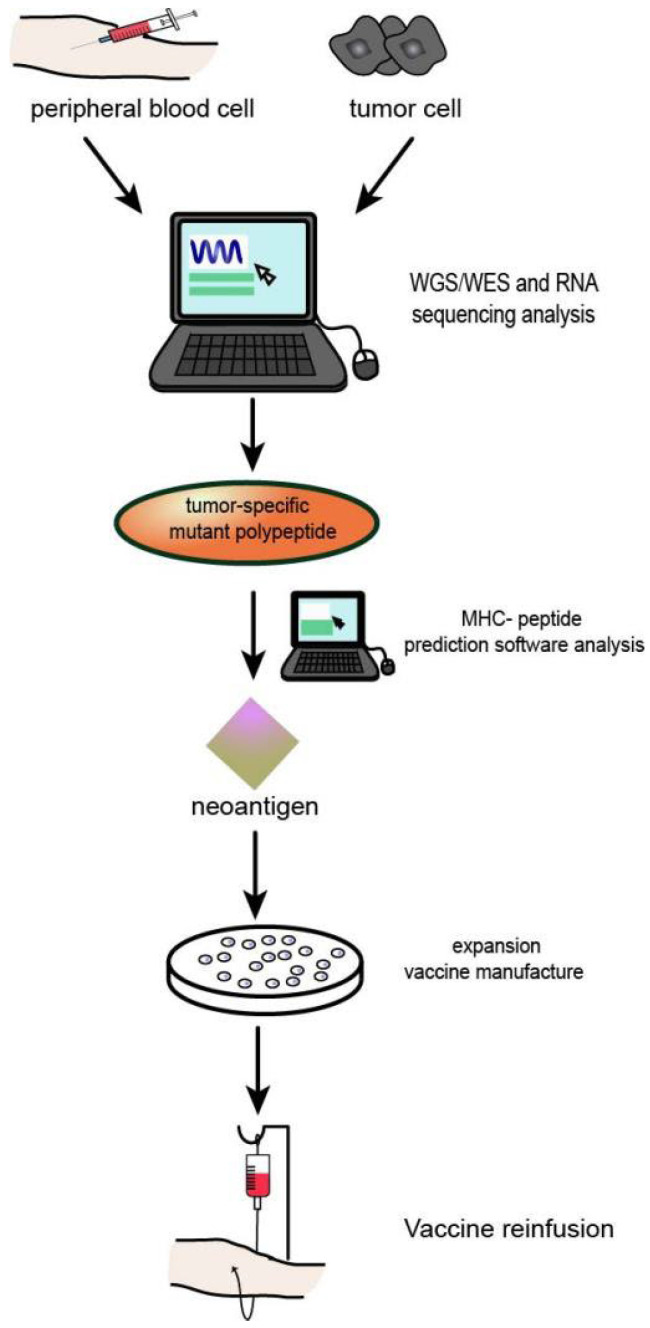
Process of neoantigen vaccine preparation: (1) Genome or whole exon sequencing was performed using tumour tissues and normal somatic cells. (2) Tumour tissue transcriptome sequencing. (3) Prediction and/or detection of mutated peptides and MHc molecular affinity. (4) Effective tumour neoantigens were synthesized into corresponding types of vaccines.

**Figure 5 vaccines-10-01272-f005:**
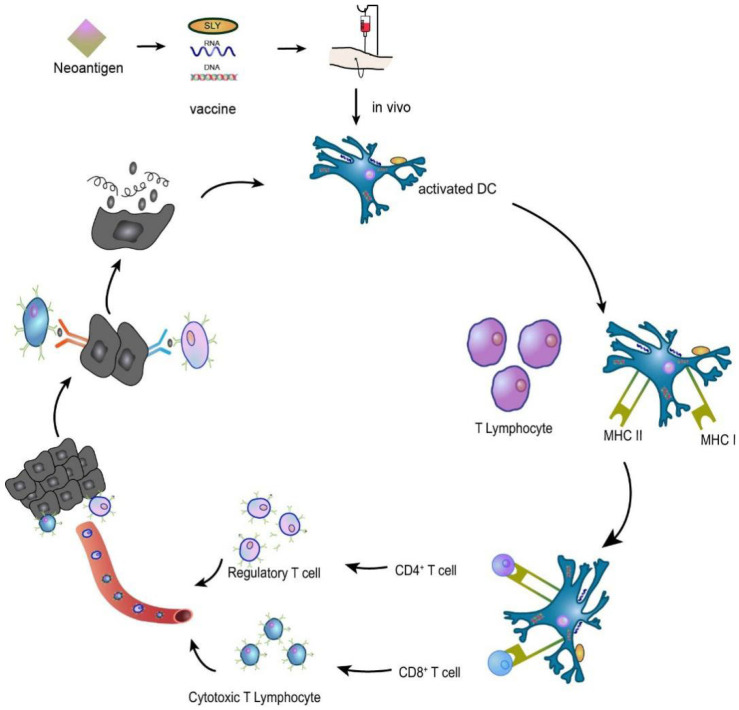
A neoantigen vaccine is introduced into a patient, prompting APCs to recognize the new antigen, process it, and present it. In vivo, neoantigen epitopes bind MHC I to promote the differentiation of CD8+ T lymphocytes into cytotoxic T lymphocytes (CTLs) that participate in the immune response or bind MHC II to activate CD4+ T lymphocytes that induce the immune response. The activated T cells migrate to the tumour site, specifically recognizing and killing the tumour cells, and the lysed tumour cells release more tumour antigens, which are subsequently presented and activated by dendritic cells (DCs), specifically killing the tumour cells and forming a virtuous cycle.

**Table 1 vaccines-10-01272-t001:** Clinical trials of neoantigen vaccines for HNSCC.

NCT Number	Study Title	Conditions	Interventions	Phase	Number Enrolled	First Posted	Inclusion Criteria
NCT03633110	Safety, Tolerability, Immunogenicity, and Antitumor Activity of GEN-009 Adjuvanted Vaccine	HNSCC, others	Biological: GEN-009 Adjuvanted Vaccine Drug: Nivolumab Drug: Pembrolizumab	I/II	24	16 August 2018	1. Patients beginning pembrolizumab with recurrent or metastatic HNSCC who experienced disease progression while on or after receiving a platinum-based therapy, or those beginning first-line pembrolizumab for recurrent or metastatic tumours. 2. Agree to a tumour biopsy 50 days after first GEN-009 vaccination.
NCT05269381	Personalized Neoantigen Peptide-based Vaccine in Combination with Pembrolizumab for the Treatment of Advanced Solid Tumours, The PNeoVCA Study	HNSCC, others	Drug: Cyclophosphamide Biological: Neoantigen Peptide Vaccine Biological: Pembrolizumab Biological: Sargramostim	I	36	8 March 2022	1. Histologically confirmed unresectable locally advanced or meta static solid malignancies. 2. Soft tissue lesion amenable for adequate tissue sampling. 3. Patients with actionable genomic abnormality including but not limited to EGFR, ALK, MET, ROS-1, RET, NTRK, KRAS or BRAF must have also received and progressed while on at least one line of prior FDA-approved targeted therapy.
NCT04266730	Trial of a Personalized and Adaptive Neoantigen Dose-adjusted Vaccine Concurrently with Pembrolizumab	HNSCC, others	Biological: PANDA-VAC Drug: Pembrolizumab	I	6	12 February 2020	1. As 1st line treatment for tumours expressing PD-L1 [Combined Positive Score (CPS) ≥ 1] as determined by an FDA-approved test. 2. As a non1st line treatment for patients with recurrent or metastatic HNSCC who experienced disease progression while on or after receiving platinum-containing chemotherapy.
NCT01998542	Safety and Tolerability Study of AlloVax^TM^ in Patients With Metastatic or Recurrent Cancer of the Head and Neck	HNSCC, HNC	Biological: AlloVax Biological: CRCL Biological: AlloStim	II	12	29 November 2013	1. Patients must have a tumour safely accessible for biopsy resulting in a minimum of 0.1 g of tumour sample for CRCL processing. 2. Patients must have visible external tumours measurable with at least one lesion deemed to be safely accessible for serial biopsy.
NCT03552718	QUILT-2.025 NANT Neoepitope Yeast Vaccine (YE-NEO-001): Adjuvant Immunotherapy Using a Personalized Neoepitope Yeast-based Vaccine to Induce T-Cell Responses In Subjects W/Previously Treated Cancers	HNSCC	Biological:YE-NEO-001	I	16	12 June 2018	1. Must have received <6 months of SoC therapy. 2. ECOG performance status of 0 to 2. 3. If cancer recurs while on treatment during this study, the patient must be willing to provide a tumour biopsy specimen for exploratory analyses, if considered safe by the Investigator.

**Table 2 vaccines-10-01272-t002:** The advantages and disadvantages of PD-1 and PD-L1 inhibitors.

	Advantage	Disadvantage
PD-1 inhibitors	1. Suitable for all types of malignant tumours. 2. Long-term efficacy. 3. Side effects are relatively minor.	1. The efficiency is not high, at less than 30%, and even lower when combined with genetic mutations. 2. The onset of action is slow, with a median of three months. 3. Expensive
PD-L1 inhibitors	1. PD-1 inhibitors only block the PD-1-PD-L1 pathway, without affecting the PD-1-PD-L2 pathway, avoiding the occurrence of interstitial pneumonia and other side effects. 2. It can block the coinhibitory function of B7.1 and PD-L1, and fully activate T cell function and cytokine production.	1. Expensive. 2. A higher dose is required for the same efficacy as PD-1 drugs.

## Data Availability

Not applicable.

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
