# Peer review of "Immunotherapeutic Strategies for Head and Neck Squamous Cell Carcinoma (HNSCC): Current Perspectives and Future Prospects"

_vaccines, 2022, doi:10.3390/vaccines10081272_

Round 1

Reviewer 1 Report

This manuscript, written Dr Gao, review type, with the title of “Immunotherapeutic strategies in head and neck squamous cell carcinoma (HNSCC): current perspectives and future prospects” focuses on neoantigens, vaccines, and immune checkpoint inhibitors in head and neck squamous cell carcinoma.

This review is well written, and it is easy to read. However, more figures and/or summary tables would make the reading experience easier. The first part of the review focused on neoantigens, but later mainly focuses on immune checkpoint inhibitors. In the abstract, neoantigens appear to be the most important, but in manuscript immune checkpoint part is quite large.

Additional comments:

 (1) Line 36. Regarding “They are often caused by high-risk human papillomavirus (HPV)”. Could you please write the specific types of HPV of high-risk? For example: “Epidemiologic and molecular studies have identified the HPV 16 genotype of HPV as a causative agent in many of these patients [1]. Other high-risk HPV genotypes, such as HPV 18, 31, or 33, are also causative but are less common. These high-risk HPV infections may also rarely cause cancers at other head and neck sites” (please paraphrase it).

 (2) In addition to tobacco and HPV infection, there are other risk factors. Could you please add “Other head and neck cancer risk factors include betel nut chewing, radiation exposure, vitamin deficiencies, periodontal disease, immunosuppression, and other environmental and occupational exposures” (please paraphrase it).

 (3) Regarding lines 45-52. Although it is not the main topic of this review. I think I figure could be made showing the mechanism of immunotherapy (immune checkpoint inhibitors such a s anti PD-L1.

 (4) Lines 58-59. Regarding “Tumours are though to be formed because of mutations, but mutations can also lead to self-destruction when tumour cells accumulate mutations to promote growth.” The second part of this sentence is difficult to understand (to me). How the mutations in oncogenes that promote cell growth leads to self-destruction?

 (5) Lines 61-63. Regarding “On the other hand, the expression of nonsynonymous mutations can produce tumour-specific antigens, and it is precisely the mutation-generated antigens that trigger T-cell antitumor immunity”. Since it is a review, it may be useful to make a note and explain the difference between synonymous and nonsynonymous mutation; and the definitions of point mutation, frame-shift mutation, nonsense mutation and missence mutation. It is well explain in the following link: https://www.differencebetween.com/what-is-the-difference-between-synonymous-and-nonsynonymous-mutation/

 (6) Line 63. Regarding “mutation-generated antigens that trigger T-cell antitumor immunity”. Could you please make a figure showing the mechanism that is explained in the lines 64-72?

 (7) Line 76. Regarding “HER2, MUC1 and MAGE”. I think that genes should be written in italics.

 (8) Lines 81-85. Regarding “tumour neoantigens are not affected by central immune tolerance”. Could you please explain with more detail why?

 (9) Lines 98-147. Regarding section 3 “Identification and Screening of Neoantigens”. Several mechanisms are being shown. If possible, I would recommend to add some figures.

 (10) Line 173. Regarding “bind to MHC class II to activate CD4+ T lymphocytes, causing them to differentiate into regulatory T cells (Tregs).” Tregs are characterized by inhibiting the host immune response. In carcinomas, usually associate to poor prognosis of the patients. Could you please confirm that the mechanisms of action of the neoantigen vaccine is to generate Tregs?

 (11) Lines 196-198. Regarding “HNSCC patients often have multiple high-frequency mutations, and thus, cancer vaccines targeting high-frequency mutations may improve antitumor efficacy and patient benefit”. Could you please specify which are the high-frequency mutations of HNSCC? Do these mutations affect HPV-related immune checkpoint or immune response markers?

 (12) Line 285. Regarding “HNSCC is usually highly immunosuppressive”. Could you please specify why?

 (13) Could you please confirm that sections 4.2, 4.2, and 4.2.1-3 are all related to neoantigens? I think that sections 4.2 are no “neoantigen restricted”.

 (14) Line 325. Could you please add that macrophages express PD-1 ligands?

 (15) There is a review in Nat Rev Cancer that may be of interest to the authors: Yarchoan M, Johnson BA 3rd, Lutz ER, et al. Targeting neoantigens to augment antitumour immunity. Nat Rev Cancer 2017; 17:209.

Reviewer 2 Report

Comments:

1. The Table 1 needs to add more details, such as which years in trials. Any specific patients' info.

2. It will benefit the audience to have a Table outlined the advantages and disadvantages for PD-1 and PD-L1 inhibitors.

3. Overall, the sample size for clinical trials is small. Any explanation.

Round 2

Reviewer 2 Report

No more comments

Author Response

Dear reviewer,

 Thanks a lot for reviewing our manuscript again. As you suggested  the English language need minor revision, we edited the manuscript again and corrected some language errors, all the changes had been marked in MS word file.

Best Regards and Yours,

Fuyuan Yang